# DR-GPT: A large language model for medical report analysis of diabetic retinopathy patients

**Joel Jaskari**[1], **Jaakko Sahlsten**[1], **Paula Summanen**[2], **Jukka Moilanen**[2],
**Erika Lehtola**[2], **Marjo Aho**[3], **Elina Säpyskä**[3], **Kustaa Hietala**[4], **Kimmo Kaski**[1,5]*

**1** Department of Computer Science, Aalto University, Espoo, Finland, **2** Department of Ophthalmology, University of Helsinki and Helsinki University Hospital, Helsinki, Finland, **3** Department of Ophthalmology, Helsinki University Hospital and University of Helsinki, Helsinki, Finland, **4** Central Finland Health Care District, Jyväskylä, Finland, **5** The Alan Turing Institute, London, United Kingdom

* kimmo.kaski@aalto.fi

**Data Availability Statement:** The patient data cannot be shared publicly because of the data protection law of Finland, the General Data Protection Regulation (GDPR) of European Union,

## Abstract

Diabetic retinopathy (DR) is a sight-threatening condition caused by diabetes. Screening programmes for DR include eye examinations, where the patient's fundi are photographed, and the findings, including DR severity, are recorded in the medical report. However, statistical analyses based on DR severity require structured labels that calls for laborious manual annotation process if the report format is unstructured. In this work, we propose a large language model DR-GPT for classification of the DR severity from unstructured medical reports. On a clinical set of medical reports, DR-GPT reaches 0.975 quadratic weighted Cohen's kappa using truncated Early Treatment Diabetic Retinopathy Study scale. When DR-GPT annotations for unlabeled data are paired with corresponding fundus images, the additional data improves image classifier performance with statistical significance. Our analysis shows that large language models can be applied for unstructured medical report databases to classify diabetic retinopathy with a variety of applications.

## Introduction

Diabetic retinopathy (DR) is a sight-threatening eye disease that develops as a result of diabetes. As a standard practice, ophthalmologists first identify signs of the disease by observation and then classify the disease based on the combination and severity of the visible signs. In Finland, the photographer and physician record the findings and resulting DR classification in the patient's electronic health records, possibly in an unstructured manner depending on the healthcare provider. In order to perform any computational analyses on a patient or population level, the DR classification label is required. However, manual labeling of retrospectively collected medical reports can become prohibitive when dealing with large scale studies involving tens of thousands of patients.

Recently, large language models (LLMs) have gained widespread popularity among the public with the rise of chat-based applications, such as OpenAI's ChatGPT, Meta's Llama, and

and our research permission granted by HUS Helsinki University Hospital that do not allow sharing of individual patients' data. The patient data are available from the HUS Helsinki University Hospital for researchers who meet the criteria for access to confidential data. More specifically the datasets are not publicly available and restriction apply to their use. The restrictions are imposed by the Wellbeing Services County of HUS Group, the joint authority for Helsinki and Uusimaa, P.o. Box 100 Fi-00029 HUS, Finland (requests for the data access can be made to tietopalvelu@hus.fi).

**Funding:** J.J. & J.S. Glaucoma Foundation Lux (https://glaukoomatukisaatiolux.fi/), Silmäsäätiö Foundation (https://www.silmäsäätiö.fi/), and Grant Number 345449 Research Council of Finland (https://www.aka.fi/en/); E.S. & M.A. Evald and Hilda Nissi Foundation (https://www.nissinsaatio.fi/index_en.htm). The funders did not play any role in the study design, data collection, analysis, decision to publish, or in the preparation of the manuscript.

**Competing interests:** The authors have declared that no competing interests exist.

Google's Bard. In the medical domain, LLMs are being developed for various classification tasks based on medical text, such as automatic extraction of significant findings in chest radiograph reports [1], disease ICD-code and treating department prediction based on patients' self-reports [2], and COVID-19 diagnosis based on chemosensory reports [3]. As such, the LLM-based approach shows promise for automatic determination of the DR classification label from unstructured text reports.

In the present study, we develop and evaluate a Finnish GPT-based system (DR-GPT) for auto-labeling the severity and gradability of diabetic retinopathy from clinician's reports with unstructured text format. We quantitatively evaluate the classification performance of the system using established measures, and additionally, we examine the utility of such system to generate weakly annotated data for training a convolutional neural network.

## Materials and methods

In this section we present the dataset used in our analyses, the data preprocessing methods, our experimental setup, and the measures used to evaluate the results. A graphical illustration of our experimental pipeline is presented in Fig 1.

### Patient data

The research was conducted as a retrospective and registry-based study using a pseudonymized dataset of diabetic patient screening and follow-up studies as well as special healthcare visits at the Helsinki University Hospital (HUS) region over a period from 2016 to 2019. The dataset consists in total of 40236 studies from 31292 patients. During each visitation, both retinal fundi of a patient were photographed, and a physician examined the fundus images to describe the visible signs and severity of the patient's DR in unstructured medical reports using the Early Treatment Diabetic Retinopathy Study (ETDRS) grading system [4]. According to the Finnish law (Medical Research Act (488/1999) and Act on Secondary Use of Health and Social Data (552/2019)) and European General Data Protection Regulation (GDPR) rules 216/679, a retrospective and registry-based study does not need ethical permission or informed consent from subjects. The research permit was granted by the Helsinki University Hospital

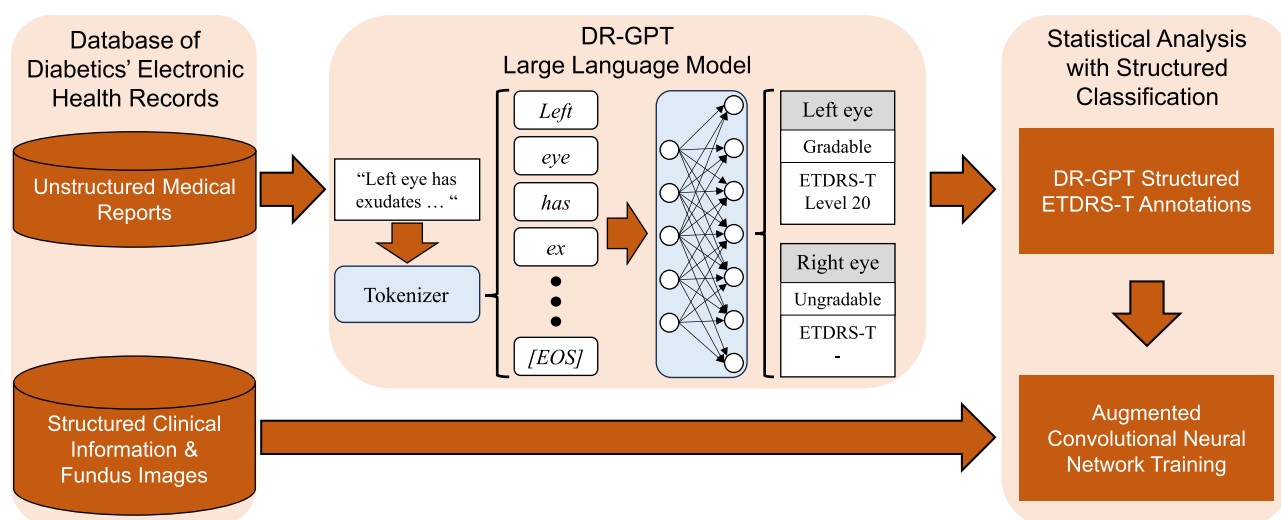

**Fig 1. Graphical illustration of our experimental pipeline for the DR-GPT large language model.**

Chief Medical Officer (decision number 67/2020), Helsinki, Finland, July 1, 2020. The data was accessed March 4, 2021 and the authors did not have access to identifying information.

For the present study, one optician, one optometrist, and two specialists in ophthalmology have manually analyzed 26626 ($\approx$ 66.2%) of the reports, such that the ETDRS severity of both the left and right eye mentioned in each report was converted to the corresponding numeric level of the system. The reports that mentioned an ETDRS severity of at most 35 were annotated by the optician or the optometrist. The cases with a higher ETDRS severity and those cases with any abnormalities, such as other pathologies or ungradable reports, were evaluated by one of the two specialists in ophthalmology. The abnormal reports were also evaluated for ambiguity of the grade, i.e., if there was ambiguity regarding the severity of DR or which eye the DR severity corresponded to, and if visible laser scars or laser treatment was mentioned. The lateral ambiguity prevents the determination of the correct label for each of the eyes, and the DR that manifests after laser treatment is not captured by the ETDRS scale because it alters the natural progression of diabetic retinopathy. The ambiguous and laser treated cases were assigned an auxiliary *Ungradable* label, while the fully gradable cases were assigned the *Gradable* label.

The dataset of patients were divided to model training, validation, and test sets with a greedy search algorithm that performed the division such that a) the reports from each patient could only reside in one of the sets, b) the distribution of ETDRS classes was to be as similar as possible in all of the sets, and c) the proportion of data in the sets was to be matched with 70%, 10%, and 20% for training, validation, and test sets, respectively. Avoiding the overlap of patients between the sets avoids overoptimistic results by the deep learning neural network memorizing possible spurious patient level patterns. The ETDRS distributions are presented in detail in the S1 Table.

We used a truncated version of the ETDRS (ETDRS-T), which includes the four least severe ETDRS classes, i.e., 10, 20, 35, and 43, and a single class that is a combination of the most severe ETDRS classes in our dataset, i.e., 47, 53A-D, 53E, 61, 65, and 71. This simplification of the grading scheme was applied to balance between the number of samples present in each class and the clinical relevance for separating class severity. Although ETDRS-T cannot differentiate the ETDRS classes $\geq$47 from one another, the cases with the ETDRS class 47 are among the first that can require treatment in the coming years, thus making this system suitable to be applied in healthcare. The class distributions of ETDRS-T in patient, report, and eye -level are presented in Table 1. In addition, we have also evaluated the DR-GPT on a binary

**Table 1. ETDRS-T distributions for the medical report data.**

| Label | Development Set* | | | Test Set | | |
|---|---|---|---|---|---|---|
| ETDRS-T / Gradable | Patients** | Reports† | Eyes | Patients** | Reports† | Eyes |
| 10 | 14124 | 15563 | 29994 | 3506 | 3885 | 7507 |
| 20 | 1658 | 1934 | 2558 | 411 | 477 | 644 |
| 35 | 823 | 1093 | 1499 | 206 | 265 | 371 |
| 43 | 303 | 475 | 706 | 69 | 118 | 180 |
| $\geq$47 | 127 | 164 | 267 | 27 | 41 | 66 |
| Gradable | 15325 | 17512 | 35024 | 4219 | 4786 | 8786 |
| Ungradable | 2998 | 3487 | 6473 | 802 | 841 | 1694 |

* Development set denotes training and validation data.

** A patient is counted for a label if the label is mentioned in any of the reports of the patient.

† A report is counted for a label if the label is mentioned for either of the eyes.

**Table 2. ETDRS-T distributions for the labelled images.**

| Label | Development Set* | | Test Set | |
|---|---|---|---|---|
| ETDRS-T | Patients** | Eyes | Patients** | Eyes |
| 10 | 14120 | 28594 | 3553 | 7210 |
| 20 | 1626 | 2441 | 405 | 622 |
| 35 | 790 | 1406 | 201 | 348 |
| 43 | 293 | 651 | 64 | 160 |
| $\geq$47 | 129 | 240 | 28 | 66 |

* Development set denotes training and validation data.

** A patient is counted for a label if the label is mentioned in any of the reports of the patient.

DR classification system (RDR) used in previous studies [5–7]. The RDR system is defined as moderate or worse diabetic retinopathy on the proposed international diabetic retinopathy classification (ICDR) system [8], with ICDR classes lower than moderate DR assigned the label 0 and moderate or worse the label 1.

We have observed that there were a number of duplicate reports in the dataset, despite them having been recorded in an unstructured manner. Specifically, there were a number of reports with identical contents for some of the patients with milder severity classes, e.g., the reports merely indicating that there was no DR to be observed. There were in total of 3818 such duplicate reports in the test set, with 3117 reports having ETDRS-T class 10 for both eyes, 351 reports with ETDRS-T class 20 for both eyes, 300 reports with ETDRS-T class 35 for both eyes, and 50 reports where there the severity was different for each eye, but at most ETDRS-T class 35. To examine the performance of DR-GPT more in depth, we formed two test datasets. One of the sets was the test data as-is, i.e., following the empirical distribution of the reports, and the other set was formed with report stratification, i.e., keeping only one example of each unique report. The DR-GPT performance on the former set measures how well it performs on the population level, whereas the latter set measures how well the DR-GPT performs on a report level.

The medical reports are based on the fundus images taken during the patient visits. In a standard visitation, four 50° field-of-view fundus photographs are taken from both eyes. The four images consist of a macula centered color image and three red-free filtered images with one of them centered at the macula, one centered temporal to the macula, and one is centered nasal to the optic disc. However, some of the patient visits include more than four images per eye due to various reasons, e.g., patient requiring additional images due to previous laser treatment or patient requiring anterior segment images to visualise optic media opacities in detail. Additionally, some of the patients have missing images, e.g., when an eye cannot be imaged due to advanced eye disease or due to technical issues during photography. For our image classification experiments we included the eye image sets with exactly four images of a standard visitation to simplify the analysis. In total, the image-based experiments included annotations for 41738 eyes of 19293 patients from 22056 visitations and 18032 eyes of 7765 patients from 10026 visitations with no annotations. We used the same training, validation, and test splits for the patients as with the report classification experiments. Full description of the ETDRS-T class distributions for the image classification experiments is shown in Table 2.

## Data preprocessing

In order to limit the number of tokens, i.e., the text representations used as input to large language models, that require processing and to prevent the DR-GPT from learning spurious

correlations between uninformative tokens and the class, we preprocessed the text data with multiple rules. The portions of the medical reports that consider observations regarding the patient's fundus are in a free-form text, whereas there are some automatically added parts that are structured. For example, the beginning and the end of the report have structured information, such as time and date of the clinical examination and the name of the examiner. Since this information does not consider the patient's health, we removed these parts from the texts. In addition, unnecessary characters added for visual purposes were automatically removed, e.g., multiple line-breaks. Finally, we utilized the text-tokenizer of TurkuNLP Finnish GPT-3 Small model [9] to convert the text strings to token indices for DR-GPT.

For our experiments with weakly annotated image data, we performed various preprocessing and data augmentation methods. The original fundus images were rectangular in shape with the fundus being visible as a circular region in the middle of the image and surrounded by black borders. We cropped each image to the smallest square image that contained the fundus entirely to remove as much of the black borders as possible. We then resized each image to a standard resolution of $512 \times 512$. During the training of the convolutional neural networks, we utilized training augmentations based on recent literature [5, 7, 10], i.e., random spatial flips both vertically and horizontally (p = 0.5), random rotations uniformly within the range of $[-180°, 180°]$, random translations within the range of [-25,25] pixels in both spatial axes, and random zooms within range [90%,110%]. Finally, the image pixel values were mapped to the range [-1,1], during both the training and inference.

## Deep learning models

As our large language model, we utilize the recently proposed Finnish GPT-3 model [9], namely the pretrained "Small" version of the architecture. The model is based on the BLOOM architecture [11], which is similar to the GPT-3, and the pretrained model has been trained in the next token prediction paradigm with various Finnish text resources. The model takes a sequence of tokens as an input and it outputs a sequence of probability distributions with the distribution at an index $i$ representing the conditional probability of the token at index $i + 1$ given all the previous tokens.

The DR-GPT model consists of two Finnish GPT-3 neural networks, one of which classifies the degree of diabetic retinopathy on the ETDRS-T scale for the left and right eyes, and the other determines the gradability of the left and right eyes. In order to adapt the model for these classification tasks, we replace the final, i.e., the next token prediction, layer of the model with two parallel layers that predict the ETDRS-T level or gradability for each eye. Additionally, when feeding the tokenized text to the model, we only utilize the predictions on the last index of the sequence due to the causal masking used within the model, which ensures that all the text data is visible for the prediction. We use the cross-entropy loss function and regularize the models with 0.2 dropout rate that turned out to have the best performance from a grid-search with the values [0.0, 0.1, . . ., 0.9]. All model parameters were fine-tuned with Adam optimizer [12] with learning rate $3 \times 10^{-6}$ that was found to be the best in the range $[1 \times 10^{-3}, 1 \times 10^{-6}]$. We also examined AdamW optimizer [13] during our hyperparameter search, but as it turned out to perform similarly to Adam, we selected the latter for our main results.

As for the image classification experiment with weakly annotated data, we utilize an ImageNet [14] pretrained EfficientNet-B6 [15] convolutional neural network (CNN). In order to enable multi-view classification based on the four retinal images, we consider a combination of the sum and maximum multi-view fusion methods, described in detail in [16]. It first feeds each of the four fundus images to the CNN to create four output vectors. The output vectors are then processed by calculating the sum and maximum features in an element-wise manner

across the four outputs. Lastly, the sum and maximum vectors are concatenated and fed to a multilayer perceptron with a softmax activation to output the conditional class probabilities. We slightly modified the original sum multi-view fusion approach by instead calculating the element-wise mean, as it turned out to be more numerically stable. We use the cross-entropy loss function, and AdamW optimizer with learning rate $3 \times 10^{-5}$, which we found to result in the best performance in the task.

## Experiments and evaluation measures

To evaluate the classification performance of the five class ETDRS-T quantitatively, we use the quadratic weighted Cohen's kappa (QWK) [17], accuracy, and balanced accuracy measures. The QWK measure has previously been used in the evaluation of the deep learning algorithms for diabetic retinopathy classification with the five-class ICDR system [7, 18, 19], whereas the accuracy and balanced accuracy are common classification evaluation measures. For the binary classification tasks of RDR and gradability classification, we use the area under the receiver operating characteristic curve (AUROC), accuracy, and balanced accuracy. We trained 10 DR-GPTs and report the mean and standard deviation of the results with both empirical and report stratified distributions of test data.

The DR-GPT model outputs two vectors of probabilities that represent the conditional distribution of the classes for the left and right eyes given the text input. To calculate QWK, accuracy, and balanced accuracy, we select the label with the maximum probability as the prediction for each eye, i.e., $\hat{y}_{left} = \arg\max_c p(y_{left} = c \mid \boldsymbol{x})$ and $\hat{y}_{right} = \arg\max_c p(y_{right} = c \mid \boldsymbol{x})$. For multi-class classification with $K$ categories, a set of $N$ ground truth labels $\{y_1, \ldots, y_N\}$, and DR-GPT predicted labels $\{\hat{y}_1, \ldots, \hat{y}_N\}$, where the laterality of the label and prediction have been omitted for clarity, the QWK is defined as follows:

$$\text{QWK} = 1 - \frac{\sum_{i=1}^{K} \sum_{j=1}^{K} (i-j)^2 C_{i,j}}{\sum_{i=1}^{K} \sum_{j=1}^{K} (i-j)^2 E_{i,j}},$$

$$C_{i,j} = \sum_{n=1}^{N} I[y_n = i] \cdot I[\hat{y}_n = j],$$

$$E_{i,j} = \frac{1}{N} \sum_{a=1}^{K} C_{i,a} \sum_{b=1}^{K} C_{b,j},$$

where $I[\cdot]$ is the indicator function, $C$ is the confusion matrix, and $E$ the expected agreement matrix. Accuracy and balanced accuracy are defined as:

$$\text{Accuracy} = \frac{1}{N} \sum_{n=1}^{N} I[y_n = \hat{y}_n],$$

$$\text{Balanced Accuracy} = \frac{1}{K} \sum_{k=1}^{K} \frac{\sum_{n=1}^{N} I[y_n = k] \cdot I[\hat{y}_n = k]}{\sum_{n=1}^{N} I[y_n = k]}.$$

To obtain the binary classification measure AUROC, one needs to evaluate the sensitivity and specificity of a classifier at multiple probability thresholds $\tau$, such that at each threshold the

predicted label is defined as:

$$\hat{y} = \begin{cases} 1 & \text{if } p(y \mid \boldsymbol{x}) > \tau, \\ 0 & \text{else.} \end{cases}$$

The area under the curve defined by the sensitivities and specificities on thresholds $\tau \in [0, 1]$ is the AUROC measure. When evaluating the AUROC of RDR classification performance (RDR AUROC), we formed the probability of RDR positive by adding the probabilities of ETDRS-T classes 35, 43, and $\geq$ 47 together.

To evaluate if there are differences between an EfficientNet-B6 model trained with manually annotated data and one trained with the data augmented with DR-GPT weak annotations, we used the 10 DR-GPTs of medical report classification experiments to generate weak annotations on the unlabelled set of 10026 reports. Specifically, we took an ensemble of the models and filtered out the reports that the ensemble DR-GPT predicted as being ungradable. For the rest of the reports, the maximum probability label was assigned as the target label. We trained 10 EfficientNet-B6 models on both manually annotated data and on manually annotated data augmented with weak annotations, and determined statistically significant differences with two-tailed Wilcoxon signed rank tests [20], by considering the p-values less than 0.05 as significant. To control the false discovery rate of multiple hypotheses, Benjamini-Hochberg procedure [21] was used to declare significant results, which accounted for the various classification measures used to compare the approaches. These classification measures were implemented in Python (3.9.12) using the scikit-learn (1.3.0) [22] and Wilcoxon signed rank tests were calculated using SciPy (1.8.0) [23].

## Results

This section presents the results for the DR-GPT on classifying patients' medical reports, and for the EfficientNet-B6 on image classification with and without the DR-GPT generated weakly supervised data.

### Classification of patient's medical reports

The results of ETDRS-T classification with the DR-GPT are presented in Table 3. On the empirical dataset, the DR-GPT reached 0.975 QWK, 0.987 accuracy, and 0.937 balanced accuracy in the ETDRS-T classification task, and 0.999 AUROC in the RDR classification. It turned out that the performance degraded slightly when evaluating the performance on the report stratified i.e., only a unique sentence, test data, the QWK value decreased to 0.962, accuracy to 0.952, and the balanced accuracy to 0.930. In addition, the RDR AUROC turned out to decrease to 0.994.

In the case of binary classification for gradability, the DR-GPT resulted in an AUROC value of 0.996, accuracy value of 0.986, and balanced accuracy value of 0.971, when evaluating the full test data. The performance on the report stratified set of patient data showed that the performance decreases slightly with AUROC being 0.960, accuracy 0.952, and balanced accuracy 0.894. The results are illustrated in full on Table 4.

**Table 3. ETDRS-T and RDR classification results of DR-GPT.**

| Test Set | QWK | Accuracy | Balanced Accuracy | RDR AUROC |
|---|---|---|---|---|
| Empirical | 0.975 (0.002) | 0.987 (0.001) | 0.937 (0.006) | 0.999 (0.000) |
| Stratified | 0.962 (0.004) | 0.952 (0.004) | 0.930 (0.006) | 0.994 (0.001) |

**Table 4. Gradability classification results of DR-GPT.**

| Test Set | AUROC | Accuracy | Balanced Accuracy |
|---|---|---|---|
| Empirical | 0.996 (0.000) | 0.986 (0.000) | 0.971 (0.003) |
| Stratified | 0.960 (0.003) | 0.952 (0.001) | 0.894 (0.013) |

In the case of the ensemble DR-GPT model, created by taking an ensemble of the 10 repetitions, the evaluation yielded 0.977 QWK, 0.988 accuracy, and 0.943 balanced accuracy in the ETDRS-T classification, and 0.999 AUROC in the RDR classification on the empirical set of test data. As for the report stratified test data, the DR classification performance resulted in 0.966 QWK, 0.957 accuracy, and 0.936 balanced accuracy in the ETDRS-T classification, and 0.995 AUROC in the RDR classification. In turn the ensemble DR-GPT model had gradability classification performance of 0.997 AUROC, 0.988 accuracy, and 0.974 balanced accuracy on the empirical test set, and 0.965 AUROC, 0.956 accuracy, and 0.903 balanced accuracy on the report stratified test set. The ensemble DR-GPT ETDRS-T and gradability confusion matrices, evaluated on the empirical set, are presented in Fig 2. The DR-GPT ensemble predictions on the unlabelled data, used in the image classification experiments, resulted in 5627 patients with 9949 eyes in the ETDTS-T class 10, 1650 patients with 2429 eyes in class 20, 1460 patients with 2808 eyes in class 35, 922 patients with 2182 eyes in class 43, and 341 patients with 664 eyes in class $\geq$ 47.

## Image classification with weak supervision

As for the image classification results with the EfficientNet-B6 CNN model, when trained with manually annotated data for ETDRS-T classification, it had the mean (standard deviation) value of 0.890 (0.005) for QWK, 0.924 (0.002) for accuracy, and 0.579 (0.030) for the balanced accuracy. In addition, when the model was evaluated for RDR classification, it had an RDR AUROC of 0.979 (0.003). It turned out that when the model was trained with the DR-GPT generated weak annotations, in addition to the manual annotations, the ETDRS-T classification performance improved to 0.892 (0.004) for QWK, 0.924 (0.003) for accuracy, and 0.604 (0.017) for balanced accuracy, in terms of the mean (standard deviation) of these measures. Furthermore, the RDR AUROC improved to 0.984 (0.001). Additionally, all the differences between the baseline supervised and the DR-GPT augmented model were statistically significant. A summary of the results are presented in Table 5 and the confusion matrices of ensembled models from both the experiments in Fig 3.

## Discussion

In this work, we have proposed a large language model DR-GPT for automatic classification of diabetic retinopathy and its gradability from unstructured medical patient reports. We demonstrated that DR-GPT has a high accuracy on both of the classification tasks, and furthermore, that it can be used to automatically annotate fundus images by analysing the corresponding unlabeled medical reports. This weakly annotated data could in turn be used to augment the training data for a convolutional neural network to improve its performance statistically significantly.

Overall the DR-GPT model had excellent accuracy with only a few errors. An observation from the analysis is that the classification performance of all grading scales degraded systematically when evaluated with the report stratified dataset in comparison to the empirical dataset. This difference can be attributed to the duplicate reports being more abundant during training

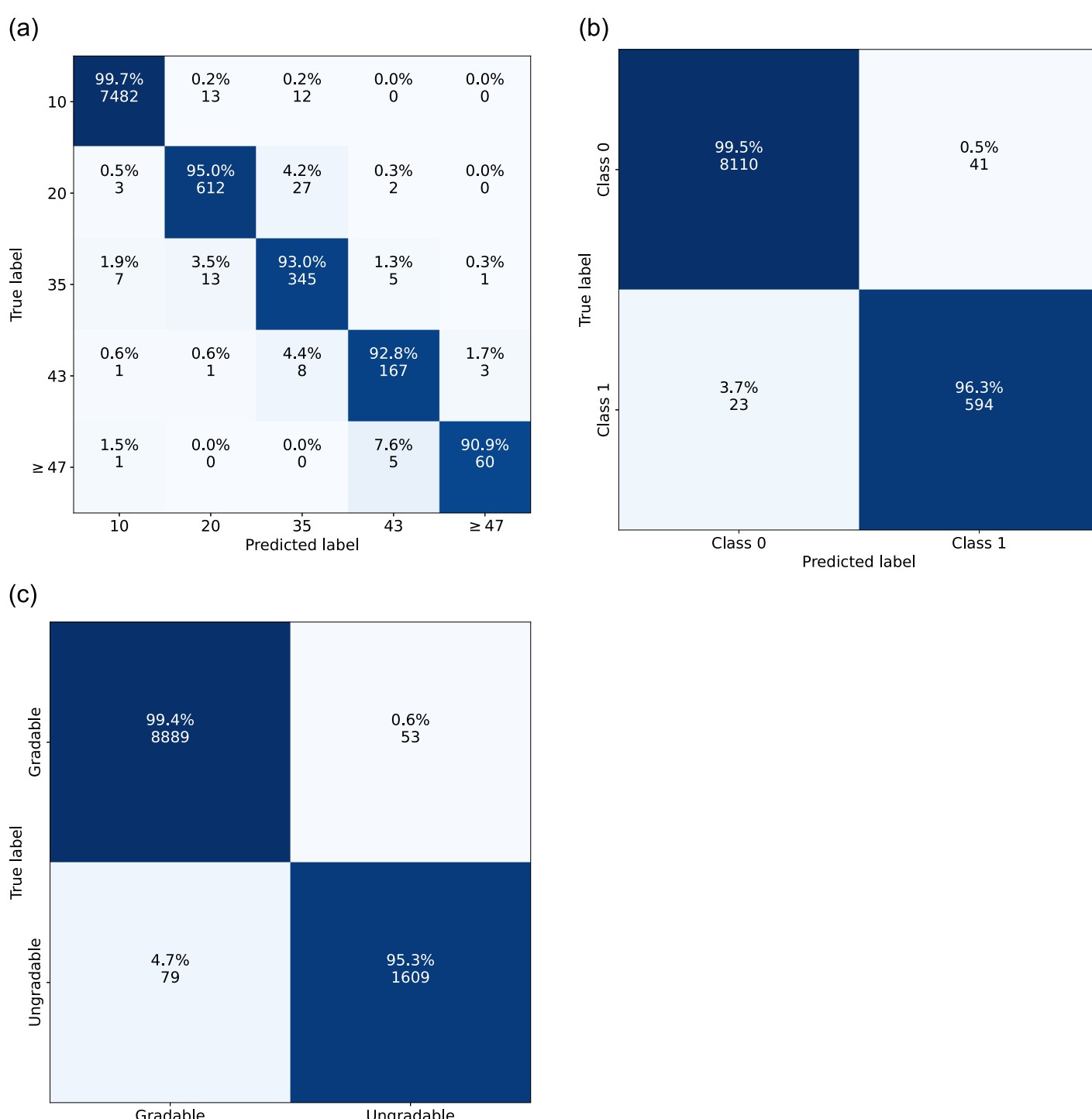

**Fig 2. Ensemble DR-GPT confusion matrices on the empirical set of test data. (a)** ETDRS-T classification, **(b)** RDR classification derived from the ETDRS-T, and **(c)** gradability classification.

**Table 5. Diabetic retinopathy classification from fundus images.**

| Experiment | QWK | p-value | Accuracy | p-value | Balanced Accuracy | p-value | RDR AUROC | p-value |
|---|---|---|---|---|---|---|---|---|
| Supervised | 0.886 (0.005) | 0.003 | 0.922 (0.002) | 0.002 | 0.642 (0.030) | 0.014 | 0.979 (0.003) | $1.8 \times 10^{-4}$ |
| + DR-GPT | 0.893 (0.004)* | | 0.926 (0.002)* | | 0.667 (0.014)* | | 0.984 (0.001)* | |

* Statistically significant differences (p < 0.05).

as well as having simpler and homogeneous content e.g., simply stating that there is no diabetic retinopathy, which makes their classification easier. Thus, the remaining performance gains for the DR-GPT could be achieved by increasing the number of the more severe cases, i.e., ETDRS-T classes 43 and $\geq 47$, in the training set, as they have more heterogeneous reports with unique terminology. Indeed, increasing the amount of training data for the severe classes can also mitigate the problem of underestimating the severity of classes 35, 43, and $\geq 47$, which is likely caused by the data-distribution being skewed towards the lower severity classes. Interestingly, the class 20 is being confused more to 35, i.e., towards more severe DR, than to 10, which would be expected due to the class-imbalance. We manually examined the cases where the class 20 was confused with 35, and discovered that out of the 27 reports misclassified, 25 reports had either inconsistencies in the reported signs and severity of the DR or were mislabeled as 20. Out of the remaining two misclassified reports, one report had a sentence describing the signs of class 35 that were present in the previous examination. This is likely the reason why the model misclassified the current DR status. The final report was misclassified by the DR-GPT without a clear explanation.

As for the image classification experiments, the EfficientNet-B6 convolutional neural network with access to additional weakly supervised data generated by DR-GPT had a significant improvement across all the measures, when compared to the supervised baseline. This demonstrates that the DR-GPT approach can be used with practically no additional cost, besides the computation, to improve the performance of an image-based classifier. However, the size of the improvement was small or moderate across the measures, even when the weakly supervised CNN had access to additional images from 19492 eyes. This outcome can be due to the saturation of classification performance with a CNN based approach, and we expect that the DR-GPT weakly supervised data could improve upon the supervised baseline more when the labelled dataset of fundus images is small. The misclassifications of the image classifier were more pronounced towards the majority, i.e., less severe, classes than the misclassifications of DR-GPT, with the exception of class 43 that the image classifier got confused more towards $\geq 47$ than to lower severity classes. These differences might be attributed to how salient the class-specific features are in the data. The medical reports should mention the visible changes in both eyes and which DR severity is assigned to each of them. While there is large variability in the terminology used in the medical reports, it is still lower than the one seen in retinal images. This is because lighting, retinal features, such as pigmentation, and other conditions of the eye introduces enormous variability in how the signs of DR are seen in the images. The variabilities can make some signs of DR less visible to the image classifier, which might lead it to underestimate the severity.

Large language models, such as DR-GPT, show promise to serve as workflow augmenting tools. The healthcare providers that only use natural language to describe the current status of patient's diabetic retinopathy could benefit from DR-GPT structuring the reports into a tabular format of severity grades without the need to develop new practices. Furthermore, as the DR-GPT classifies also the gradability of DR, it might be beneficial for quality control of

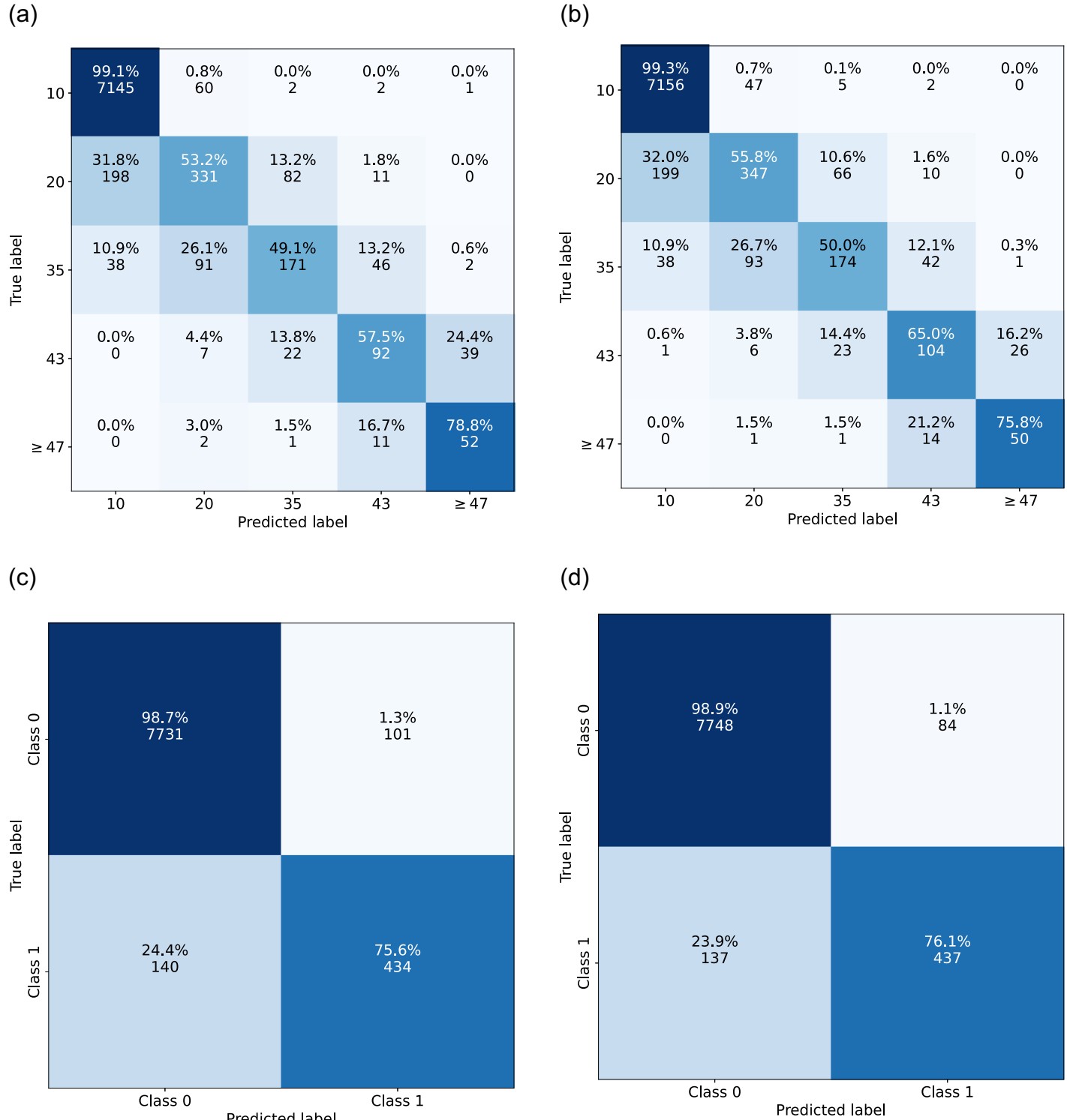

**Fig 3. ETDRS-T and RDR confusion matrices for image classification with ensembles. (a)** ETDRS-T with standard supervised approach, **(b)** ETDRS-T with DR-GPT weakly supervised data, **(c)** RDR with standard supervised approach, **(d)** RDR with DR-GPT weakly supervised data.

medical reports. DR-GPT might have utility for DR screening purposes, due to it having systematically better accuracy the lower the DR grade is. However, we emphasize that as a medical professional needs to always be responsible for the treatment and monitoring of patients, the DR classifications by DR-GPT should not be used as-is for treatment planning without the accompanying report and supervision by a responsible practitioner. Moreover, the DR-GPT could be utilised as a tool to gather annotated data for image-based analyses. Indeed, once trained, the DR-GPT could serve as a cost-efficient method to label a database of medical reports that have the corresponding images available. This circumvents the need for expensive and time-consuming manual annotation practice.

In the context of previous work, DR-GPT classifies diabetic retinopathy with the most detailed classification scale, i.e., the ETDRS-T. In [24], a long short-term memory (LSTM) -based natural language processing system was developed to detect the presence of various conditions in medical reports. That system could also classify DR, but only with a binary DR or no-DR classification scale. In contrast, DR-GPT classifies DR with the clinically more relevant five grade ETDRS-T categories that differentiate various stages of development of DR. Furthermore, the DR-GPT predictions can be aggregated to the simple DR or no-DR system. In this case, it achieves in F1-score, precision, and recall measures a value over 98%, whereas the LSTM system has correspondingly a value less than or equal to 78%. Orthogonal to our work, [25] proposed a large language model -based approach to identify DR related concepts and their relations in the medical reports. Specifically, the task was to discover mentions of lesions, and their laterality, severity, and location. As the DR-GPT was trained to identify the severity and gradability of DR, and the aforementioned LLM was trained to identify the concept-relation patterns of DR related terms, the language models serve a different purpose. In the medical practice, both models could work simultaneously, as DR-GPT could classify the severity of DR and the LLM of [25] the concepts that were observable in the report. However, as the main focus of this study is to examine DR classification, the evaluation of such concept-relation LLM is out of the scope of this study.

There are some limitations to our study. Firstly, the DR-GPT was trained using exclusively Helsinki University Hospital region retrospective data with medical reports in Finnish language, which may differ in terms of how structured the medical patient reports are in comparison to other national regions. This poses a challenge in the generalization of the DR-GPT to reports from other regions, datasets, and languages, all of which can be analyzed further with multi-center data. While our findings provide evidence for the technical validity of the DR-GPT, a prospective analysis should be considered to provide clinical evidence of the method prior to real-world usage. Secondly, there are medical reports that refer to the previous examinations that the patient has undergone, which were not available to the DR-GPT during training or inference. This lack of information could be remedied by architectural and data-processing development, which remains for the future work. Thirdly, we utilized the truncated ETDRS-T system instead of the ETDRS scale due to the limited number of cases with severe or worse background DR and proliferative DR in our dataset, which limits the applicability of the approach to more fine-grained analysis of DR. Lastly, our CNN experiments with DR-GPT generated weakly supervised data utilised a relatively simple multi-view fusion mechanism, which limited our analysis to the standard cases with exactly four images.

This pivotal study has shown the efficacy of LLMs with unstructured medical data and the synergy in combination with CNNs for image -based analysis for diabetic retinopathy screening. With the promise of the high accuracy of the method, more robust analysis of prospective, multi-language, and multi-center data is warranted prior to clinical use. In the future, larger labelled datasets, specifically with more severe cases of DR, would facilitate the use of finer diagnostic grading scales.

## Conclusion

A large language model can accurately analyze diabetics' unstructured medical reports to identify the severity and gradability of diabetic retinopathy. It can be used to automatically generate structured diabetic retinopathy classification for existing databases and to generate training data for other deep learning -based models.

## Supporting information

**S1 Table. ETDRS distributions for the medical report data.** A patient is counted for a label if the label is mentioned in any of the reports of the patient, and a report is counted for a label if the label is mentioned for either of the eyes. Development set denotes training and validation data.
(XLSX)

## Author Contributions

**Conceptualization:** Joel Jaskari, Jaakko Sahlsten, Paula Summanen, Jukka Moilanen, Kustaa Hietala, Kimmo Kaski.

**Data curation:** Paula Summanen, Erika Lehtola, Marjo Aho, Elina Säpyskä.

**Funding acquisition:** Paula Summanen.

**Investigation:** Joel Jaskari, Jaakko Sahlsten, Kimmo Kaski.

**Methodology:** Joel Jaskari, Jaakko Sahlsten.

**Project administration:** Paula Summanen, Jukka Moilanen, Erika Lehtola, Marjo Aho, Elina Säpyskä, Kustaa Hietala, Kimmo Kaski.

**Software:** Joel Jaskari, Jaakko Sahlsten.

**Supervision:** Paula Summanen, Jukka Moilanen, Kimmo Kaski.

**Validation:** Paula Summanen, Kimmo Kaski.

**Visualization:** Joel Jaskari, Jaakko Sahlsten.

**Writing – original draft:** Joel Jaskari, Jaakko Sahlsten.

**Writing – review & editing:** Joel Jaskari, Jaakko Sahlsten, Paula Summanen, Jukka Moilanen, Erika Lehtola, Marjo Aho, Elina Säpyskä, Kustaa Hietala, Kimmo Kaski.

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
