## [Decision Letter · Decision Letter 0]

7 Aug 2024

PONE-D-24-00877DR-GPT: a large language model for medical report analysis of diabetic retinopathy patientsPLOS ONE

Dear Dr. Kaski,

Thank you for submitting your manuscript to PLOS ONE. After careful consideration, we feel that it has merit but does not fully meet PLOS ONE’s publication criteria as it currently stands. Therefore, we invite you to submit a revised version of the manuscript that addresses the points raised during the review process.

We look forward to receiving your revised manuscript.

Kind regards,

Ditta Zobor, MD

Academic Editor

PLOS ONE

3. In the online submission form, you indicated that your data is available only on request from a third party. Please note that your Data Availability Statement is currently missing the name of the third party contact or institution / contact details for the third party, such as an email address or a link to where data requests can be made. Please update your statement with the missing information.

Reviewers' comments:

Reviewer's Responses to Questions

**Comments to the Author**

1. Is the manuscript technically sound, and do the data support the conclusions?

Reviewer #1: Yes

Reviewer #2: Yes

2. Has the statistical analysis been performed appropriately and rigorously? 

Reviewer #1: I Don't Know

Reviewer #2: Yes

3. Have the authors made all data underlying the findings in their manuscript fully available?

Reviewer #1: No

Reviewer #2: Yes

4. Is the manuscript presented in an intelligible fashion and written in standard English?

Reviewer #1: Yes

Reviewer #2: Yes

5. Review Comments to the Author

Reviewer #1: In their study titled "DR-GPT: a large language model for medical report analysis of diabetic retinopathy patients", Jaskari et al. present DR-GPT, a large language model (LLM) designed to classify the severity of diabetic retinopathy (DR) from unstructured medical reports. DR-GPT is based on pre-trained Finnish GPT models and fine-tuned for the classification and gradeability tasks. The study examines the efficacy of DR-GPT annotated data in improving the performance of image-based classifiers when combined with manually labeled data, demonstrating statistically significant enhancements.

The study is well-structured, and the methodology appears robust, with a clear explanation of the data preprocessing, model training, and evaluation processes. The automatic analysis of diabetic retinopathy from medical records is a clinically relevant topic, which is typically a manual and time-consuming process. The paper is easy to read. The figures and tables effectively support their respective content.

Minor comments:

Discussion

- Overall, the discussion would be more informative if the authors contextualized their results by discussing relevant publications in the field (e.g., doi: 10.1186/s12911-022-01996-2).

- What is the authors' assessment of the potential for real-world clinical implementation of the DR-GPT model? The discussion would benefit from addressing the potential impact of incorrect classifications on patient care, as well as ethical and legal aspects of responsibility and governance. Specifically, how might DR-GPT be integrated into existing DR screening workflows? What are the potential benefits and challenges of implementing such a system in clinical practice?

- Although the authors only demonstrated a moderate enhancement of image classifiers, it is worth considering how this potential could be leveraged to achieve a significant benefit from the use of DR-GPT annotated images.

- The authors should discuss why they believe DR-GPT has more under-classified cases than over-classified cases in terms of DR severity.

- Although important limitations have been included, such as the limited generalizability to other healthcare systems, it should also be noted that the study's generalizability may be limited to other languages and healthcare systems due to its focus on a single language (Finnish).

- ll. 25-29: This information should be moved to the discussion and methods sections, respectively, rather than being included in the introduction.

Overall, while LLMs have been applied to medical texts before, developing a model specifically for diabetic retinopathy reports and evaluating its utility for weak monitoring is novel. The investigation of this use case and its combination with image classifiers is relevant and makes an original contribution.

The manuscript would be a valuable addition to the literature with the suggested revisions incorporated. I thank the authors for their great work.

Reviewer #2: This research applies LLM for automatic classification of DR reports. Automatic annotation of unstructured medical reports by DR-GPT reduces the workload of manual annotation and improves the efficiency, which is an innovative point with practical application prospects, and demonstrating the potential of cross-modal data combination.

Majors:

While the model performs well on specific datasets, its ability to generalize across different datasets and diverse clinical scenarios needs to be assessed. It will be good to validate further with data from different countries or healthcare systems.

Clinical evidence: It will be good to validate DR-GPT in prospective or real-world scenario, instead of just use retrospective dataset validation.

Utility scenarios: Will the DR images annotated by DR-GPT be used for patient report? Or train DL models further? Maybe it’s good to consider further experiments targeting these scenarios to show the reliability of DR-GPT.

6. PLOS authors have the option to publish the peer review history of their article (what does this mean?). If published, this will include your full peer review and any attached files.

Reviewer #1: No

Reviewer #2: No

---

## [Author Response · Author response to Decision Letter 0]

29 Aug 2024

Response to Reviewers’ comments

We thank the Reviewers for thorough reading of our manuscript and constructive feedback for improving the manuscript. Changes made to the revised manuscript based on the comments by Reviewer #1, Reviewer #2, or both are marked in yellow, green, and cyan respectively, as well the corresponding line rows of the revised text is noted. Our responses to point-by-point remarks below are in blue.

Reviewer #1: In their study titled "DR-GPT: a large language model for medical report analysis of diabetic retinopathy patients", Jaskari et al. present DR-GPT, a large language model (LLM) designed to classify the severity of diabetic retinopathy (DR) from unstructured medical reports. DR-GPT is based on pre-trained Finnish GPT models and fine-tuned for the classification and gradeability tasks. The study examines the efficacy of DR-GPT annotated data in improving the performance of image-based classifiers when combined with manually labeled data, demonstrating statistically significant enhancements.

The study is well-structured, and the methodology appears robust, with a clear explanation of the data preprocessing, model training, and evaluation processes. The automatic analysis of diabetic retinopathy from medical records is a clinically relevant topic, which is typically a manual and time-consuming process. The paper is easy to read. The figures and tables effectively support their respective content.

Our response: We want to thank the reviewer for expressing interest in our manuscript, and for the thorough analysis and comments of our manuscript.

Minor comments:

Discussion

- Overall, the discussion would be more informative if the authors contextualized their results by discussing relevant publications in the field (e.g., doi: 10.1186/s12911-022-01996-2).

Our response: We have now added discussion about the relevant publications in the field and how our study positions among these works. The relevant publications that we found were Saban et al. (doi:10.1177/19322968241228555) that was published after our initial submission date, and Yu et al. (doi:10.1186/s12911-022-01996-2); which we thank the reviewer for pointing out. The paragraph related to these studies is found in the Discussion (ll. 316-334). 

- What is the authors' assessment of the potential for real-world clinical implementation of the DR-GPT model? The discussion would benefit from addressing the potential impact of incorrect classifications on patient care, as well as ethical and legal aspects of responsibility and governance. Specifically, how might DR-GPT be integrated into existing DR screening workflows? What are the potential benefits and challenges of implementing such a system in clinical practice?

Our response: We thank the reviewer for this important comment. We consider AI-based methods as augmenting tools that can be integrated into physicians' workflow to enhance their efficiency and overall clinical practices. A system such as DR-GPT could e.g., automatically provide a structured diabetic retinopathy labelling that can be recorded into a database without the need to develop additional user-interfaces for the physicians to use, or DR-GPT could provide an alternative opinion on classification based on the findings in a medical report. As DR-GPT has better performance on the less severe cases, which are also less challenging to analyse by physicians, it could also be suitable for initial diabetic retinopathy screening purposes. However, we emphasise that the responsibility on patient care should remain on the acting physician, and systems, such as DR-GPT, should not be used fully autonomously. Indeed, a fully automated system would require safeguards for misclassification and additional training to understand the limitations of these methods e.g., expected error rate, which should be set by the governing healthcare institution. We have amended the manuscript to address these points in the Discussion section (ll. 301-311).

- Although the authors only demonstrated a moderate enhancement of image classifiers, it is worth considering how this potential could be leveraged to achieve a significant benefit from the use of DR-GPT annotated images.

Our response: We consider that DR-GPT can be also used to provide weak annotations for image classifier pretraining, which can be later fine-tuned with accurate labels more efficiently i.e., by reducing the required human annotated labels, while the only direct cost of DR-GPT is for the uptime required for the computer it runs on. This consideration is added to the manuscript in the Discussion section (ll. 311-315). 

- The authors should discuss why they believe DR-GPT has more under-classified cases than over-classified cases in terms of DR severity.

Our response: We agree with the need for additional analysis and we have now discussed this phenomenon in the Discussion section in case of the DR-GPT and the image classifier. We have also included discussion about possible factors that account to the differing classification performances between both models for DR severity classification (ll. 268-279, 290-300). 

- Although important limitations have been included, such as the limited generalizability to other healthcare systems, it should also be noted that the study's generalizability may be limited to other languages and healthcare systems due to its focus on a single language (Finnish).

Our response: We agree with this remark and we have now added the use of a single language and dataset as a limitation in the Discussion (ll. 336-337, 339-340, 354-356).

- ll. 25-29: This information should be moved to the discussion and methods sections, respectively, rather than being included in the introduction.

Our response: We have moved the main findings from the Introductions as requested and revised our main findings in the Discussion section (ll. 254-258). Moreover, our description of the method is moved to the Methods section (ll. 28-29). In addition, we have modified the last paragraph in the Introduction to better clarify the novel contributions of our manuscript (ll. 19-25).

Overall, while LLMs have been applied to medical texts before, developing a model specifically for diabetic retinopathy reports and evaluating its utility for weak monitoring is novel. The investigation of this use case and its combination with image classifiers is relevant and makes an original contribution.

The manuscript would be a valuable addition to the literature with the suggested revisions incorporated. I thank the authors for their great work.

Our response: We thank the reviewer for the positive and encouraging comments that we believe have improved our manuscript now that we have addressed them all. 

Reviewer #2: This research applies LLM for automatic classification of DR reports. Automatic annotation of unstructured medical reports by DR-GPT reduces the workload of manual annotation and improves the efficiency, which is an innovative point with practical application prospects, and demonstrating the potential of cross-modal data combination.

Our response: We thank the reviewer for the time spent in reviewing our manuscript, and for the suggestions made to improve it.

Majors:

While the model performs well on specific datasets, its ability to generalize across different datasets and diverse clinical scenarios needs to be assessed. It will be good to validate further with data from different countries or healthcare systems.

Our response: We agree with this remark. However, our research permit only allows for the analysis of data gathered in 2016-2019 in the Helsinki University Hospital (HUS) region in Finland. Our work serves as a first study to the detailed classification, with ETDRS-T system, of diabetic retinopathy from medical reports, and further investigation with multi-national and multi-institutional data is well warranted in the future. We have now highlighted the limitations due to single language and institution in the Discussion (ll. 336-337, 339-340, 354-356).

Clinical evidence: It will be good to validate DR-GPT in prospective or real-world scenario, instead of just use retrospective dataset validation.

Our response: We thank the reviewer for this comment. We agree that a prospective validation is critical prior to the utilisation of DR-GPT algorithm in healthcare. However, as we pointed out previously, our research permit only allows for the use of the retrospective data. We consider our study as an important milestone towards justifying the use of prospective medical report data of diabetic retinopathy, which in turn would enable more accurate evaluation of DR-GPT in the future. We have amended the manuscript to reflect this aspect in the Discussion (ll. 336-337, 340-342).

Utility scenarios: Will the DR images annotated by DR-GPT be used for patient report? Or train DL models further? Maybe it’s good to consider further experiments targeting these scenarios to show the reliability of DR-GPT.

Our response: We thank the reviewer for raising this point. Indeed, DR-GPT would require more extensive prospective validation before implementation to healthcare systems, and as such, could not be used in the current form for patient reports in clinical practice. However, as we showed in our experiments, DR-GPT produced annotations can be used to train other DL models for image classification. Indeed, the results in Table 5 demonstrate that when DR-GPT provides additional training data for a convolutional neural network, its performance improves with statistical significance. Moreover, the DR-GPT approach could be extended to full ETDRS scale in the future pending larger quantities of labelled data especially with diabetic retinopathy severities for classes above 47. This consideration is added in the Discussion (ll. 356-357).

---

## [Decision Letter · Decision Letter 1]

20 Sep 2024

DR-GPT: a large language model for medical report analysis of diabetic retinopathy patients

PONE-D-24-00877R1

Dear Dr. Kaski,

We’re pleased to inform you that your manuscript has been judged scientifically suitable for publication and will be formally accepted for publication once it meets all outstanding technical requirements.

Kind regards,

Ditta Zobor, MD

Academic Editor

PLOS ONE

Reviewers' comments:

Reviewer's Responses to Questions

**Comments to the Author**

1. If the authors have adequately addressed your comments raised in a previous round of review and you feel that this manuscript is now acceptable for publication, you may indicate that here to bypass the “Comments to the Author” section, enter your conflict of interest statement in the “Confidential to Editor” section, and submit your "Accept" recommendation.

Reviewer #1: All comments have been addressed

Reviewer #2: (No Response)

2. Is the manuscript technically sound, and do the data support the conclusions?

Reviewer #1: Yes

Reviewer #2: (No Response)

3. Has the statistical analysis been performed appropriately and rigorously? 

Reviewer #1: I Don't Know

Reviewer #2: (No Response)

4. Have the authors made all data underlying the findings in their manuscript fully available?

Reviewer #1: No

Reviewer #2: (No Response)

5. Is the manuscript presented in an intelligible fashion and written in standard English?

Reviewer #1: Yes

Reviewer #2: (No Response)

6. Review Comments to the Author

Reviewer #1: Thank you for taking the time to address the comments I made on your previous submission. I have reviewed the updated version, and I am pleased to see that you haveve made the necessary changes.

Reviewer #2: (No Response)

7. PLOS authors have the option to publish the peer review history of their article (what does this mean?). If published, this will include your full peer review and any attached files.

Reviewer #1: No

Reviewer #2: No

---

## [Editor Report · Acceptance letter]

30 Sep 2024

PONE-D-24-00877R1 

PLOS ONE

Dear Dr. Kaski, 

I'm pleased to inform you that your manuscript has been deemed suitable for publication in PLOS ONE. Congratulations! Your manuscript is now being handed over to our production team.

Kind regards, 

on behalf of

Dr. Ditta Zobor 

Academic Editor

PLOS ONE